

# Low-Noise Permalloy Ring-Cores for Fluxgate Magnetometers

David M Miles[1,2], Miroslaw Ciurzynski[1], David Barona[1], B. Barry Narod[3], John R Bennest[4], Andy Kale[1], Marc Lessard[5], David K. Milling[1], Joshua Larson[2], Ian R Mann[1]

[1]Department of Physics, University of Alberta, Edmonton, AB, Canada
[2]Department of Physics and Astronomy, University of Iowa, Iowa City, IA, USA
[3]Department of Earth, Ocean and Atmospheric Sciences, University of British Columbia, Vancouver, BC, Canada
[4]Bennest Enterprises Ltd., Summerland, BC, Canada
[5]Department of Physics and Astronomy, University of New Hampshire, Durham, NH, USA

*Correspondence to*: David M Miles (david-miles@uiowa.edu)

**Abstract.**

Fluxgate magnetometers are important tools for geophysics and space physics providing high precision magnetic field measurements. Fluxgate magnetometer noise performance is typically limited by a ferromagnetic element that is periodically forced into magnetic saturation to modulate, or gate, the local magnetic field. The parameters that control the intrinsic magnetic
noise of the ferromagnetic element remain poorly understood. Much of the basic research into producing low-noise fluxgate sensors was completed in the 1960s for military purposes and was never publicly released. Many modern fluxgates depend on legacy Infinetics S1000 ring-cores that have been out of production since 1996 and for which there is no published manufacturing process. We present a manufacturing approach that can consistently produce fluxgate ring-cores with a noise of ~6–11 pT per square root Hertz—comparable to many of the legacy Infinetics ring-cores used worldwide today. As a result, we demonstrate that we have
developed the capacity to produce the low-noise ring-cores essential for high-quality, science-grade fluxgate instrumentation. This work has also revealed potential avenues for further improving performance, and further research into low-noise magnetic materials and fluxgate magnetometer sensors is underway.

## 1 Introduction

Fluxgate magnetometers (e.g., Primdahl, 1979) are important tools for geophysics, solar-terrestrial and space physics, space
exploration, and for the monitoring space weather. They provide high-precision measurements of Earth's magnetic field that can be used to image downward into the Earth, resolving subsurface conductivity via magnetotellurics, and upward into near-Earth space, inferring the currents and waves coupling the ionosphere to the magnetosphere. Fluxgate magnetometers deliver a magnetic field measurement through modulating (gating) the local magnetic field by periodically saturating a piece of ferromagnetic core material—often in the form of a ring (Figure 1). The ferromagnetic core alone would act as a magnetic-flux concentrator but
combined with a toroidal drive winding to periodically drive it into magnetic saturation it acts as a magnetic-flux modulator - or a fluxgate. The addition of a solenoidal sense winding completes the fluxgate sensor as the, now modulated, magnetic field induces a current or voltage which can be conditioned and digitised.



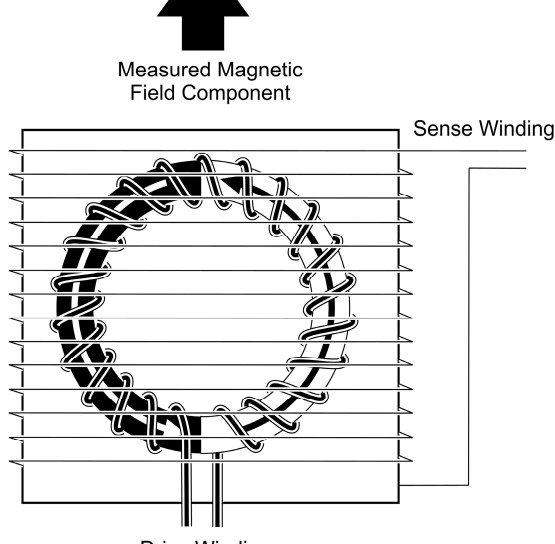

**Figure 1: A fluxgate sensor has three primary components: a ferromagnetic core, a drive winding to periodically force the ferromagnetic core into magnetic saturation, and a sense-winding to pick up the modulated (gated) field. Reproduced from Miles et al., (2017).**

The instrumental noise floor of the sensor is typically limited by the intrinsic magnetic noise of the ring-core as it is driven in and out of magnetic saturation. Despite their widespread use, the parameters that control the limiting intrinsic magnetic noise of a fluxgate ring-core remain poorly understood. The key research and manufacturing process for low-noise fluxgate ring-cores are insufficiently documented (Narod, 2014) for comparable ring-cores to be reproduced. Here we document a ring-core manufacturing process that has been developed from the limited historical documentation that exists, much of which dates to the 1960s. This

process yields fluxgate ring-cores comparable to those produced historically and serves as a baseline for further low-noise fluxgate ring-core development.

### 1.1 History of 6-81 Permalloy

A preferred ferromagnetic material used in fluxgate sensors is 6-81 permalloy containing 6% molybdenum, 81.3% nickel, and the remainder iron. The 6-81 permalloy is visible in the ring-core in Figure 2 as the glossy grey metal within the black supporting

bobbin. The earliest known reference to 6-81 permalloy is by Odani (1964) who examined magnetic properties for 5.3–6.8% molybdenum permalloy processed into thin foils and heat treated. Pfeifer (1966) undertook similar but wider ranging work that was introduced to the English language literature by English and Chin (1967). Pfeifer and Boll (1969) explored magnetic properties of similar alloys for applications such as transformers and magnetic amplifiers. The US Naval Ordinance Laboratory was aware of the utility of 6-81 permalloy as early at 1965 (Scanlon, 1966); however, few details of their research are available to the public.



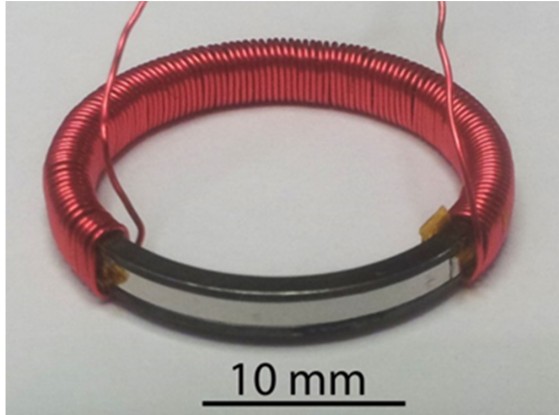

**Figure 2: An S1000 compatible fluxgate ring-core. The red enamelled wire forming the drive winding has been partially removed for the photograph exposing the ferromagnetic sense element (silver) and the supporting bobbin (black).**

Geyger (1962a, 1962b) suggested the use of thin foils as a way to suppress eddy currents when constructing fluxgate

magnetometers. However, it seems unlikely that, at the time, the researchers understood how the choice of foil thickness was impacting parameters such as grain size that appear to have controlled the magnetic noise of the sensors they were constructing (e.g., later work by Pfeifer and Kunz, 1977; Pfeifer and Radeloff, 1980). The potential of 6-81 permalloy to be used in magnetic field instruments was established in a seminal work by Gordon et al. (1968). The approach to sensor construction presented here is based on this historical research and attempts to develop a process which can produce comparable results.

**1.2 The S1000 Ring-Core**

Somewhat remarkably, virtually all the permalloy used in North American fluxgate magnetometers appears to have been manufactured from a single batch of 6-81 permalloy, likely by the Hamilton Watch Company, in or around 1969. This permalloy was then rolled into 12-micron and 3-micron thick foils and was subsequently processed to engineer low magnetic noise by two groups—the US Naval Surface Weapons Center (NSWC) White Oak (now a Department of Agriculture facility) and Infinetics

Inc. (Scarzello et al., 2001). Many modern fluxgates depend on legacy Infinetics S1000 ring-cores that have been out of production since 1996. Infinetics ring-cores are in use in terrestrial applications such as the US EarthScope (Schultz, 2009), the Canadian NRCan magnetic observatory array (Danskin et al., 2009), the Geospace Observatory (GO) Canada CARISMA magnetometer array (Mann et al., 2008); and on space missions including US MAGSAT sensor (Acuña et al., 1978), the UK Double Star fluxgate sensor (Carr et al., 2005), and Canadian e-POP MGF sensor (Wallis et al., 2015). From this historical context, it becomes apparent

why many magnetometers in use world-wide have nearly identical sensors (cf. Figure 3); they all appear to be using legacy Infinetics S1000 ring-cores. For this reason, the ring-cores developed under this project were designed to be compatible with the S1000 form-factor.





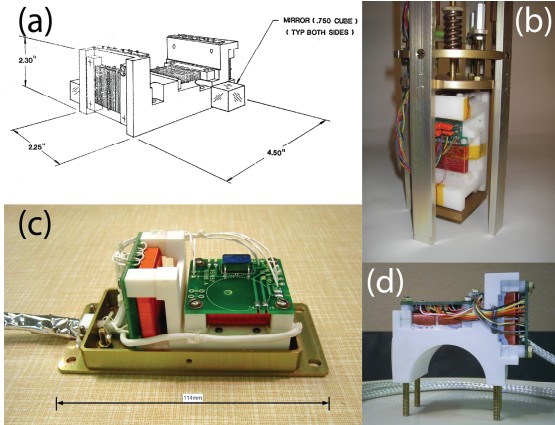

**Figure 3: Example fluxgate sensors believed to be based on the Infinetics S1000 ring-core. (a) US MAGSAT sensor (image reproduced from Acuña et al., 1978), (b) Terrestrial NIMS magnetometer used in the NSF Earth Scope project (Narod and Bennest, 1990), (c) the UK Double Star fluxgate sensor (image reproduced from Carr et al., 2005), and (d) Canadian e-POP MGF sensor (image reproduced from Wallis et al., 2015).**

## 3 Construction and Heat Treatment of New Ring-Cores

The physical construction of the fluxgate ring-core consisted of several steps illustrated in Figure 4. A circular bobbin (a) established the geometry of the ring-core and provided mechanical strength. The ferromagnetic element, used to modulate the magnetic field in the sensor, was constructed by spiral winding a thin strip of cold-rolled permalloy foil coated with an insulator (b). The assembled bobbin and strips were heat treated in a reducing atmosphere to optimize their magnetic properties (c). The ring-core was then electrically isolated using polyimide tape (d) and a toroidal drive winding was applied (e).





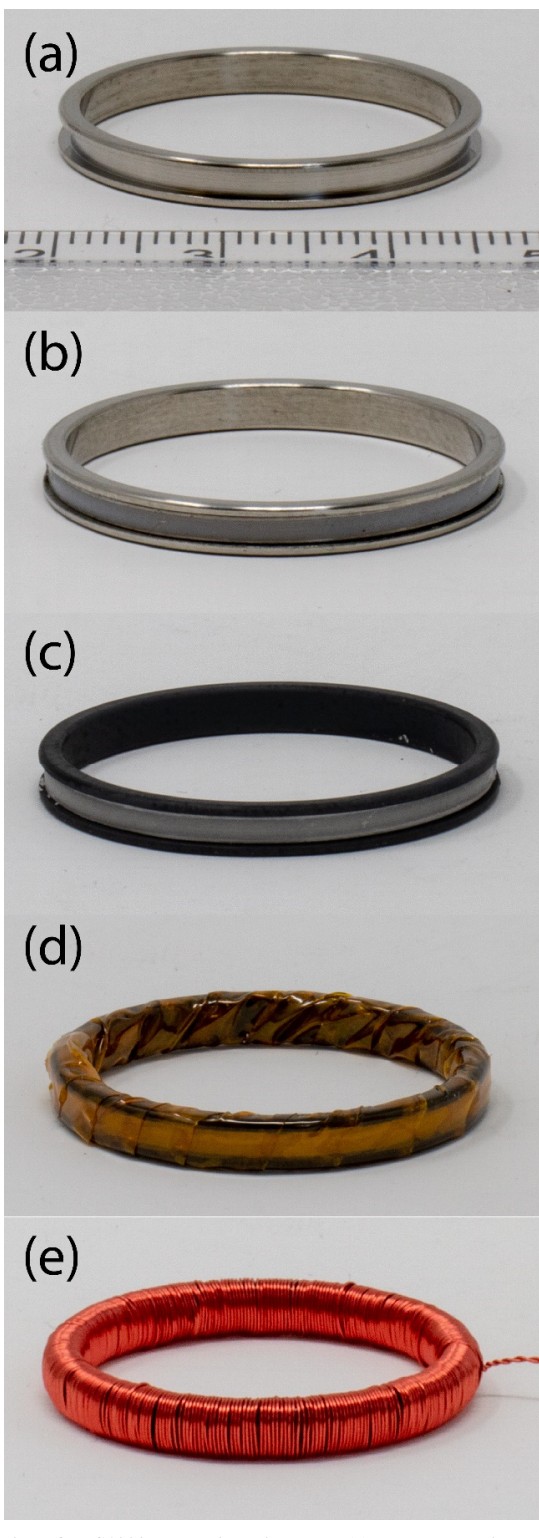

**Figure 4: Major steps in the production of an S1000 compatible ring-core. (a) Inconel bobbin used to form the ring-core and provide mechanical support. (b) Insulator coated permalloy foil strips spiral wound and welded to bobbin. (c) Same, but after heat treatment. (d) Wrapped in polyimide tape. (e) Drive winding applied.**



### 3.1 Ring-Core Bobbin

The bobbins in the ring-cores described here had the geometry of the common S1000 bobbin whose dimensions are shown in Figure 5. The bobbin defined the geometry of the ring-core and provided mechanical support to prevent the permalloy from experiencing mechanical strain after heat treatment. Even minor deformations or stresses applied to the ferromagnetic element,

such as when pushing on the ring-core to turn it within the sense winding, were found to significantly increase magnetic noise. The ferromagnetic element was spiral wound into a groove machined into the outer circumference of the bobbin. This allowed a toroidal drive winding to be subsequently wound onto the bobbin without contacting or imparting stress onto the permalloy strip.

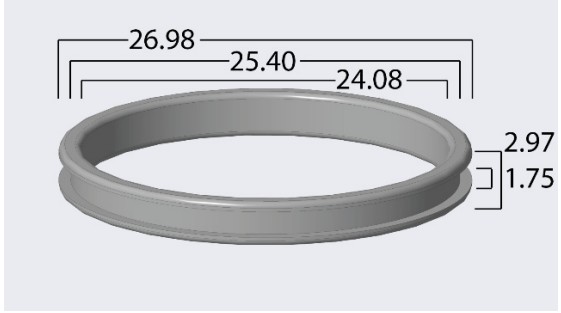

Figure 5: Schematic of a bobbin compatible with the S1000 geometry (units in mm).

The bobbins were manufactured from Inconel x750 that was selected as being non-magnetic and providing high rigidity even at the elevated temperatures of the heat treatment required to optimize the magnetic properties of the ring-core. However, Inconel x750 was a imperfect match to the permalloy sense element in terms of linear thermal expansion (12.6 ppm °C$^{-1}$ for Inconel x750, and estimated to be about 11.6 ppm °C$^{-1}$ for 6-81 permalloy). Properties for the Inconel x750 were taken from the Special Metals Group of Companies data sheet, Unified Numbering System for Metals and Alloys, reference UNS N07750. The immediate impact

of the thermal mismatch was differential expansion during the heat treatment leading to a loose fit of the permalloy strip on the bobbin in the final ring-core assembly. The differential expansion may also have enhanced the magnetic noise by introducing mechanical stress during the heat treatment and if the final ring-core assembly was operated over a wide temperature range. This effect has not yet been investigated in detail. For future designs, alternative bobbin materials that are a closer thermal match are being explored.

### 3.2 Manufacturing Thin 6-81 Permalloy Foil

A custom 4 kg ingot of 6-81 permalloy was created using a vacuum arc furnace to create a 50–50 alloy of molybdenum and nickel and then melting in the remaining constituents in a conventional furnace. The complete alloy was intended to be reduced by hot-rolling. However, the process was stopped almost immediately as the alloy began to develop severe surface cracking (Figure 6). This may have resulted from partial melting near the surface due to inhomogeneity in the mix or the alloy being non-eutectic. The

cooled ingot was instead machined into 3 mm thick stock using a milling machine. The 3 mm stock was then heat treated at 1100 °C in a 5% hydrogen atmosphere for seven days to homogenize the material before further processing.


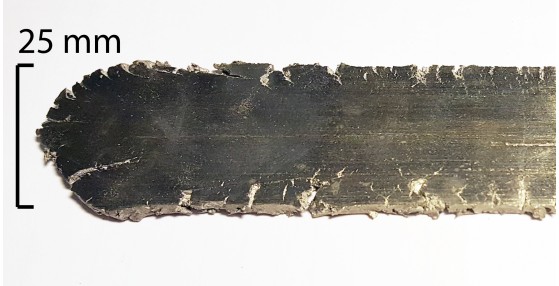

**Figure 6: 3 mm thick stock machined from the 6-81 permalloy ingot showing the surface cracking that resulted from an attempt at hot-rolling.**

Successive cold-rolling reduced the permalloy from 3 mm to the final foil thickness of ~100 μm in the ring-cores presented here.

Cold-rolling was intended to lock mechanical strain into the permalloy that the authors speculate serves as a free energy source to drive crystal grain growth during the final heat treatment, in a process referred to now as primary recrystallization (Pfeifer and Radeloff, 1980). For this reason, no interim heat treatments to soften the material were performed during the cold-rolling. The material became harder as it was thinned and strained during each cold-roll (work hardening) requiring many small passes each accomplishing a reduction of less than ~5% of the thickness. The hardness of the cold rolled permalloy and the desired thickness

(≤100 μm) placed significant requirements on the rolling mill. Deformation of the rollers and/or compression of the bearings created a practical limit of ~100 μm for the current rolling configuration.

The thinnest results to date were achieved using a Cavallin Bench Rolling Mill for Plate/Strip Model L.80/44-044 adapted with a reducing gear and a 735 W electric motor. The 44 mm diameter rollers and solid bronze bushings allow more force to be applied to the foil than other, nominally comparable, rolling mills. The current rolling mill setup (Figure 7a) reduced 3 mm permalloy

stock (Figure 7b) to 100 μm foil (Figure 7c) in approximately 250 passes. Subsequent passes provided no further significant reductions in the foil thickness.

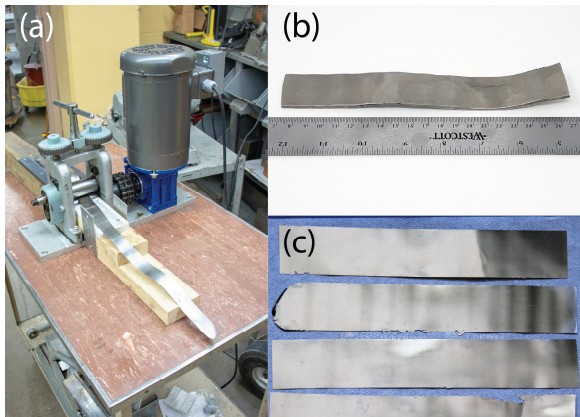

**Figure 7: (a) Modified Cavallin rolling mill used to reduce the permalloy foil. (b) 3 mm thick slice of a custom 6-81 permalloy ingot. (c) Cold-rolled 100 μm thick foil.**

The permalloy was then cut into 1.57 mm wide strips to fit within the groove cut into the external face of the bobbin used in S1000 form factor ring-cores (cf. Figure 5). The cold rolled permalloy, despite work hardening, remained sufficiently ductile that it tended to fold rather than cut. A sharp guillotine shear could cut the permalloy foil if supported by a sacrificial brass sheet but electro-discharge machining (EDM) and water jetting, while requiring more expensive infrastructure, were found to provide superior cutting results.


The Permalloy strips were coated with magnesium oxide (an electrical insulator) to prevent the formation of spot welds between layers when the strip was attached by electrical discharge welding, and to prevent the tightly-wound layers from fusing with each other during heat treatments. This insulator was created using an established process (Bill Billingsley Sr, personal communication) from milk of magnesia, $Mg(OH)_2$ diluted with water to reduce its viscosity and form a consistent thin layer. The strip was dipped,

hung up, and allowed to air dry (Figure 8a). During the heat treatment the milk of magnesia residue (Figure 8b) formed a thin but robust layer of magnesium oxide $MgO$ that electrically isolates each layer and minimises eddy currents.

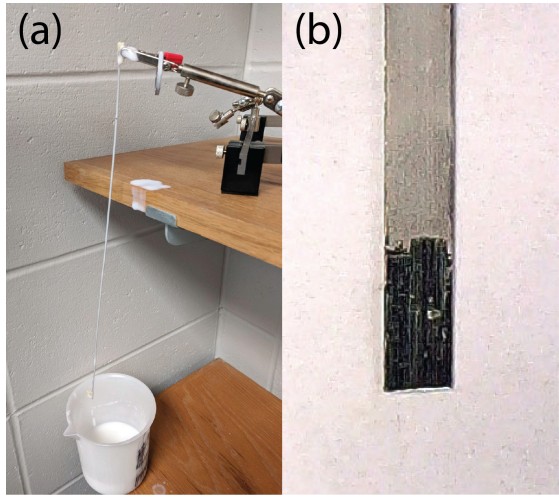

**Figure 8: (a) Oxide layer is formed by dip coating with milk of magnesia which forms magnesium oxide during heat treatment. (b) Closeup of a strip of cold-rolled permalloy strip showing milk of magnesia coating and cleared region for welding.**

**3.3 Manufacturing a Ring-core Assembly**

The ferromagnetic element of the ring-core was made up of insulated permalloy strip spirally-wound onto the Inconel bobbin. Depending on the foil available some sensors used one continuous strip while others used several strips welded together to be long enough for six turns on the Inconel bobbin (Figure 9a). The end of the permalloy strip was spot welded to the bottom of the channel cut into the outer circumference of the bobbin. The strip was cut to length such that the start and end of the strip were aligned. The

ends of the permalloy strip created an unwanted magnetic asymmetry in the ring-core that manifested in the output of sensor. Aligning the start and end of the strip localised this asymmetry allowing the sensor output to be tuned for maximum symmetry by rotating the ring-core within the sense winding. In the case of double-wound sensors, this procedure could be carried out for the first (inner) sense winding prior to affixing the second (outer) sense winding. The spiral winding was terminated by scraping away a small amount of the oxide layer (Figure 9b) and spot-welding the end of the strip down to the layer immediately underneath

(Figure 9c). The 50-cm-long Permalloy strip used in a 6-layer core had resistance of about 1.8 ohm. After spot welding such a strip on a bobbin, the resistance between the outer spot weld and the bobbin was lower than 0.5 ohm, indicating that there were more than two spot welds within the core. The consequences of these additional welds are unclear and alternative methods of affixing the foil are being investigated.



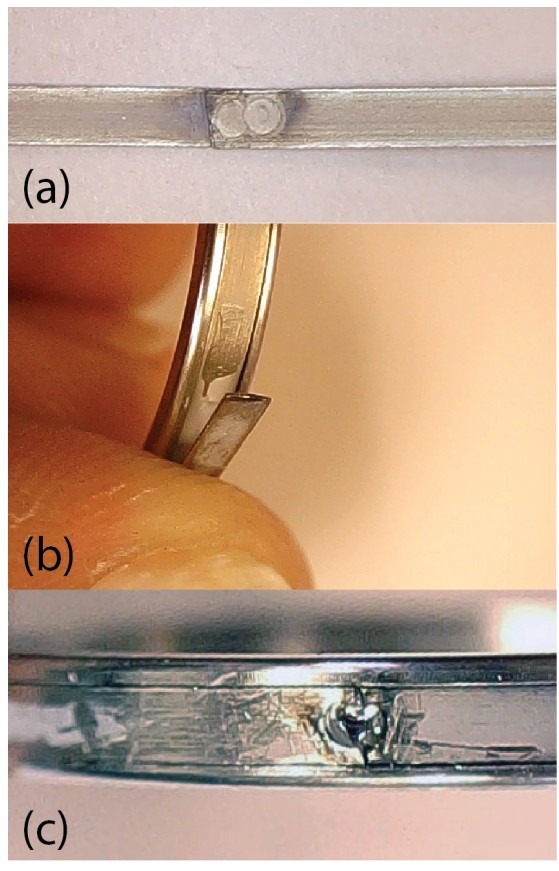

**Figure 9: (a) Permalloy strips joined by spot welding. (b) Permalloy strip wound onto bobbin ready for final weld (note patch of removed magnesia coating). (c) Permalloy strip spot-welded to itself to finish the spiral wind.**

### 3.4 Heat treatment of the Ring-core Assembly

The assembled fluxgate ring-cores were heat treated to produce high-permeability, low-coercivity, and repeatable re-magnetization properties in the ferromagnetic material that the authors hypothesize produced a relatively stress-free crystalline structure and therefore low magnetic noise. This approach was guided by the theory of the origin of fluxgate magnetic noise developed in Narod (2014). The heat treatment was intended to develop the largest possible grains in the given thickness of the permalloy foil without developing undesirable fabric where the easy axes directions are misaligned with respect to the desired magnetizing direction (e.g.,

Major and Martin, 1970; Odani, 1964).  Figure 10a shows a generic temperature profile for a ring-core showing a rapid heating from ambient temperature (A-B), a high-temperature dwell (B-C), a controlled ramp down (C-D), and a slow ramp through the disordering range and down to ambient temperature (D-E). The furnace had a maximum thermal slew rate due to its heating power, thermal mass, and potential for thermal-shock failure of the work-tube that prevented it from rapidly heating from ambient to the dwell temperature. To circumvent this, the furnace was programmed to execute a slower temperature increase and once the furnace

had reached the desired peak dwell temperature, the ring-cores were rapidly inserted into the hot-zone using a mobile loading-plate as illustrated in Figure 12. Rapid heating had the effect of maximizing grain growth due to primary recrystallization. A factor of as much as ten times larger average grain size volume is achievable by rapid compared to slow heating.



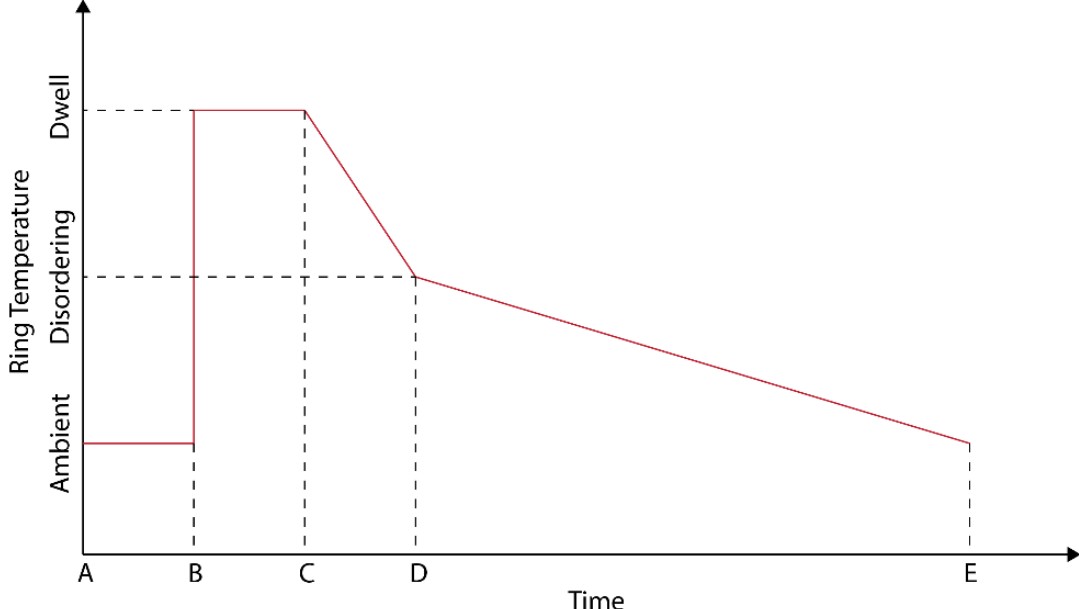

**Figure 10: Example heat-profile for a ring-core showing a rapid heating from ambient temperature (A-B), a high-temperature dwell (B-C), a controlled ramp down (C-D), and a slow ramp through the disordering range and down to ambient temperature (D-E).**

At the dwell temperature, the mechanical stress embedded in cold-rolled permalloy strips is believed to enhance primary

recrystallization of the heat-treated material. The ring-cores were heat treated in a protective (oxygen-free) atmosphere produced by a gas mixture of 5% hydrogen and 95% argon continuously injected at 100 mL/min. The hydrogen was intended to prevent the formation of oxides and to react with and remove light element impurities in the permalloy strips. The hydrogen was diluted in argon to mitigate the risk of explosion.

The heat-profile shown in Figure 10 was drawn loosely from that given by Gordon et al. (1968). The dwell temperature and duration

were determined empirically based on test ring-cores constructed from 100 μm 6-81 permalloy strip, processed at dwell temperatures in the range of 1100 to 1200 °C, and subsequent measurements of their magnetic noise. The ideal dwell temperature and duration would enable complete primary recrystallization but minimise secondary recrystallisation where a small number of grains grow at the expense of many other primary recrystallised grains. This grain growth depends on the thickness of the permalloy strip and hence would not be expected to be the same for the 100 μm strip used and the historical 3–12 μm strips. The optimum

temperature was not well-constrained; performance over a wide range of dwell temperatures has not yet been assessed in detail and is the subject of ongoing work. The C-D cooling rate is thought to be non-critical and was completed as a furnace cool (i.e., the fastest rate that could be safely executed by the furnace). In this case, -35 °C per hour was used to reduce the risk of cracking to the ceramic work-tube. Gordon et al. (1968) defined the critical ordering range for 6-81 permalloy to be from 600 to 400 °C. The D-E cooling rate was -22.5 °C per hour compared to -35 °C per hour in Gordon et al. (1968). The impact of this slower cooling

rate on the magnetic noise of the ring-cores has not yet been explored.

**3.5 Heat-Treatment Process Furnace Set-up**

Ring-core heat-treatment was undertaken using a modified Carbolite-Gero STF160 tube furnace configured for operation with a hydrogen atmosphere. The furnace was adapted, as shown schematically in Figure 10a, by the addition of a loading chamber where ring-cores could be staged at room temperature while the furnace heated and then rapidly inserted into the hot-zone without



violating the controlled atmosphere. A complete heat treatment system set up consisted of a furnace, gas cabinet, ring-core loading chamber, transport system and a data acquisition system. Figure 11 shows a diagram and a photograph of the heat-treatment system used at the University of Alberta.

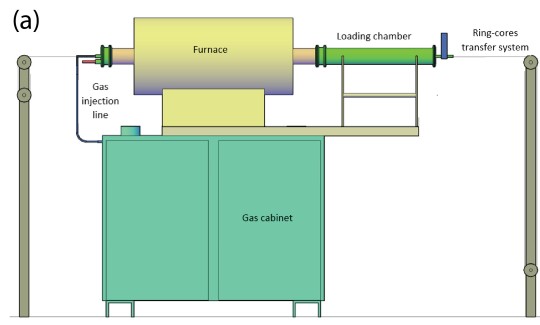

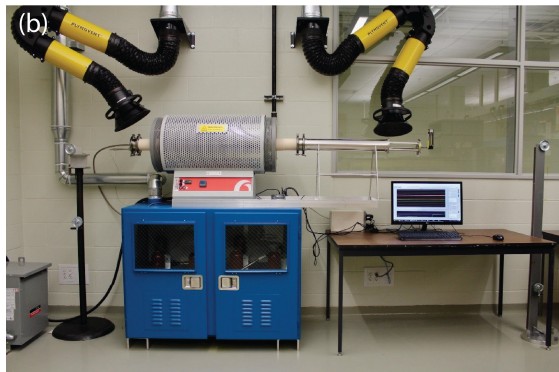

5      **Figure 11: Diagram (a) and photograph (b) of the heat-treatment system showing loading chamber and wire-pulley used for hot-insertion.**

The furnace hot-zone comprised a work-tube of high-purity (99.8%) alumina that is impermeable and inert to hydrogen. The adaptation for rapid ring-core insertion is shown schematically in Figure 12. A high-purity alumina Dee-tube (Coorstek, AD-998) was inserted into the alumina work-tube to create a flat surface on which parts could move. The heat plug at the end of the work-

10   tube adjacent to the loading chamber was modified to create an opening aligned with the surface of the Dee-tube through which the ring-cores could be inserted. In the loading chamber, aluminium tubes coplanar with the surface of the Dee-tube created the staging area for the ring-cores.



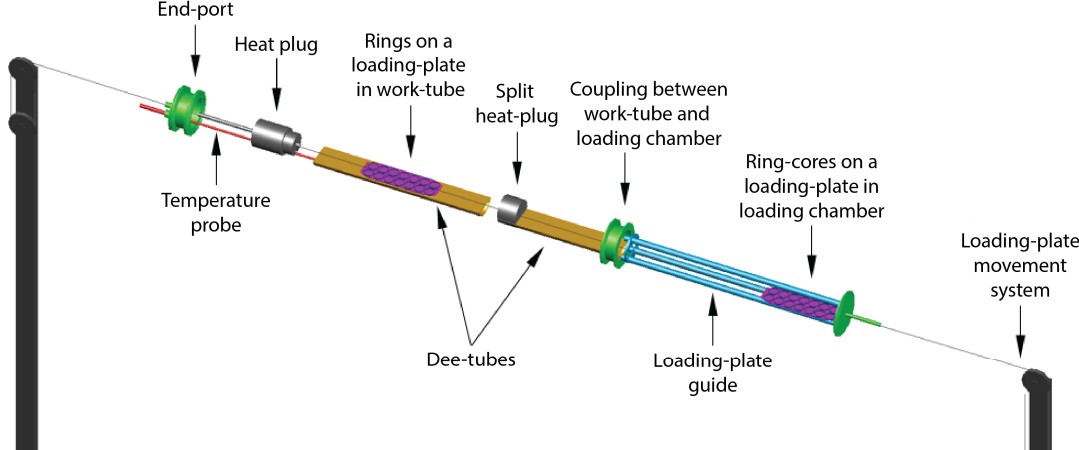

**Figure 12: Exploded view of the internal components of the furnace and the loading chamber. Ring-cores were staged in the loading chamber for rapid-insertion after the furnace reached the nominal dwell temperature.**

The ring-cores were placed on a molybdenum loading-plate (Figure 13) that could be pulled into the hot-zone using tungsten wire

5   threaded through the length of the work-tube. Alumina powder was used to prevent the ring-cores from welding to the loading-plate during the heat treatment.

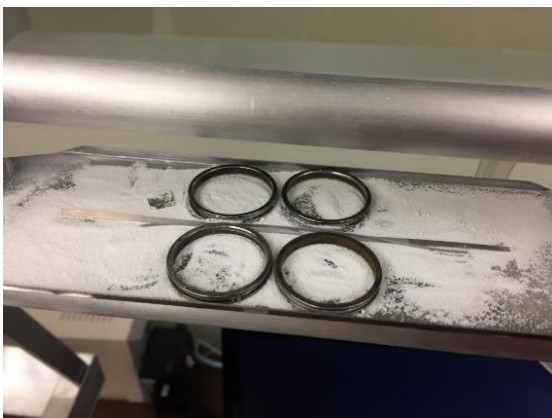

**Figure 13: Ring-cores on a molybdenum loading-plate in a loading chamber. Alumina powder prevented the ring-cores from welding to the plate.**

10   **4 Effect of Heat Treatment on Permalloy Foil Strips**

The heat treatment had significant effects on the crystalline structure of permalloy strips as shown in Figure 14. The width (horizontal dimension) of the permalloy strip in each photograph is about 1.57 mm. The correlation between grain size and the magnetic noise of permalloy ring-cores in fluxgate sensors continues to be investigated. Larger grains are believed to produce less magnetic noise but only if they grow through primary recrystallization (Herzer, 1997). Further research on this topic is needed.



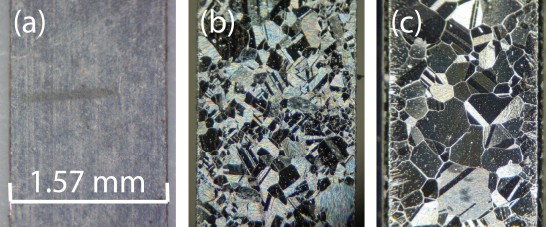

**Figure 14: Optical image of permalloy grains (a) before heat treatment and after heat treatment at (b) 1150 °C and (c) 1200 °C.**

The effect of the heat treatment of the magnetic properties of the permalloy was tested by examining the B-H curves of heat-treated test-strips. The strips were cyclically re-magnetized by supplying a 1 kHz sinusoidal voltage to a drive winding. Each magnetization cycle was captured in 250k readings of magnetizing current and coil-induced voltage. B values were derived from dB/dt values through numerical integration and are shown in arbitrary units (AU). Figure 15 shows the effect of a heat treatment on B-H curves of a single, straight permalloy strip (100 × 1.57 × 0.1 mm) excited at 1 kHz and averaged over 400 magnetisation cycles.

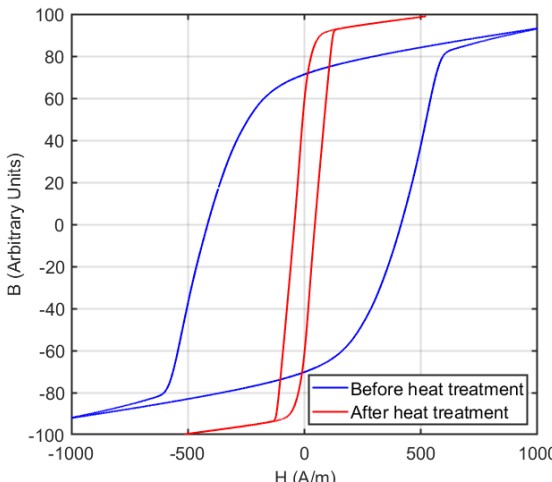

**Figure 15: B-H loop of permalloy ring-cores before (blue) and after (red) heat treatment.**

The heat treatment reduced coercivity of the permalloy strip by a factor of about six. It seems likely that heat treatments can be further optimised in terms of dwell temperature, dwell time, and cooling rates; this is still being investigated. The ideal and optimised thermal profile for heat treatment is also likely to depend on the permalloy thickness.

**5 Ring-core Characterization**

The new ring-cores were integrated into a fully working fluxgate magnetometer to assess their noise performance. Each ring-core was given a toroidal drive-winding of ~232 turns of 32 AWG enamelled wire to drive the ferromagnetic element of the ring-cores into periodic magnetic saturation. The wound ring-cores were each mounted on a small printed circuit boards for efficient and repeatable testing. Each ring-core was rotated within a 360 turn sense winding to minimize the coupling of the drive signal into the sensor output. Figure 16 shows the custom fixture used for testing the ring-cores.



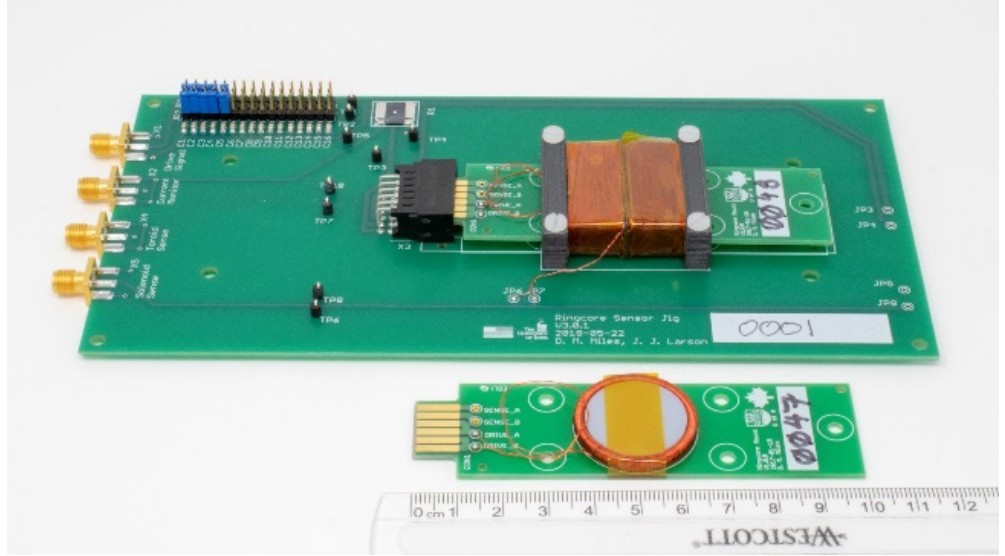

**Figure 16: Test jig used to rapidly characterize multiple ring-cores. Each ring-core has a toroidal drive winding applied and is mounted on a small printed circuit board. The mounted ring-core inserts through a modified sense winding and the entire assembly is placed in a solenoid/magnetic shield for testing.**

5   The fluxgate ring-cores were characterised by driving and interrogating them with a fluxgate magnetometer electronics package based on a heritage design (Miles et al., 2013, 2016; Narod and Bennest, 1990; Wallis et al., 2015) utilizing second-harmonic tuned drive with direct digitisation (Figure 17a), which had been generalized to drive and interrogate a wide range of ring-cores. Figure 17b shows a single-axis laboratory fluxgate electronics design built for this project. The magnetometer operated on external benchtop voltage supplies allowing the ring-core drive signal to be rapidly optimised for different ring-core designs. The sequence

10  of large power inductors, visible on the right half of the circuit board, combined with a similar capacitor bank on the sensor fixture and custom firmware allowed the drive circuit to be rapidly tuned to achieve the rapid and deep magnetic saturation required for low-noise operation.





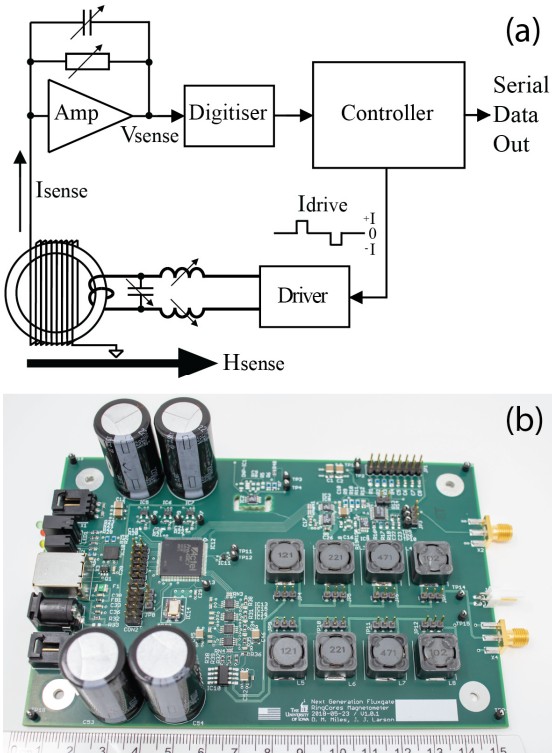

**Figure 17: Simplified functional diagram (a) and photograph (b) of one fluxgate magnetometer channel with configurable drive and preamplifier.**

The ring-cores were driven at 5 kHz and tuned to create large amplitude and short duration current spikes (Figure 18a). The sense winding was used in a short-circuit current-output configuration creating waveforms at the output of the pre-amplifier as shown in Figure 18b. The ambient magnetic field created second harmonic modulation that was sampled using synchronous digitisation indicated by the vertical dashed red lines in the figure.





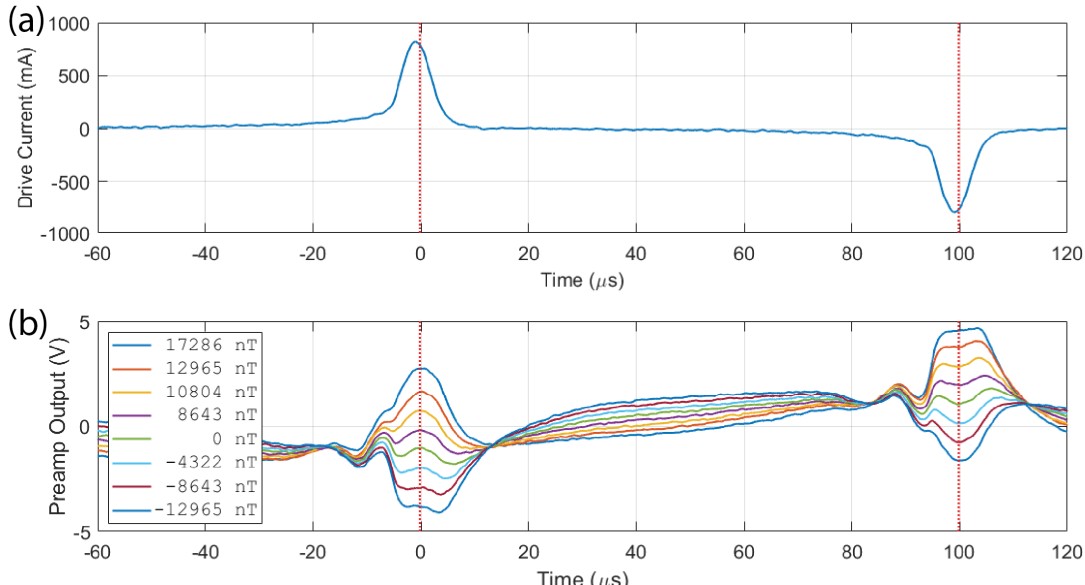

**Figure 18: (a) Current wave from tuned drive circuit ensure the ring-cores are in deep magnetic saturation. (b) Timeseries of the sense-coil output, after preamplifier, showing response to different magnetic field intensities.**

## 6 Performance of the Ring-Cores

The ring-cores were characterized for noise performance, aging effects, and for basic thermal and vibration flight qualification. The noise floor of each ring-core was calculated by calibrating the coupled ring/sensor/electronics against a known magnetic field using a solenoid inside a magnetic shield. The solenoid was then switched off and 20 minutes of magnetically quiet data were recorded and processed to produce power spectral density (PSD) using a unit-correct quantitative implementation of Welch's method (Heinzel et al., 2002; Welch, 1967) with a 2048 point Fast Fourier Transform (FFT), a 1536 point overlap, and a Hann window function. The figure of merit for each ring-core was taken to be the value at 1 Hz of a line fit to the low frequency (1/f) region of the quiet data PSD, giving a quantitative estimate of the magnetic noise at 1 Hz.

Twenty-two ring-cores were manufactured using the process described herein and heat treated in three batches of 2, 10 and 10. The average power spectral noise density was very similar in ring-cores from all three batches with variability decreasing slightly in the final batch likely because of improved consistency in spot welding of the permalloy strips. Figure 19 shows a histogram for the noise level of these ring-cores compared to those for historical 3 µm and 12 µm Infinetics ring-cores. The noise measurements were performed in a three-layer mumetal magnetic shield with a residual field strength of ~50 nT.



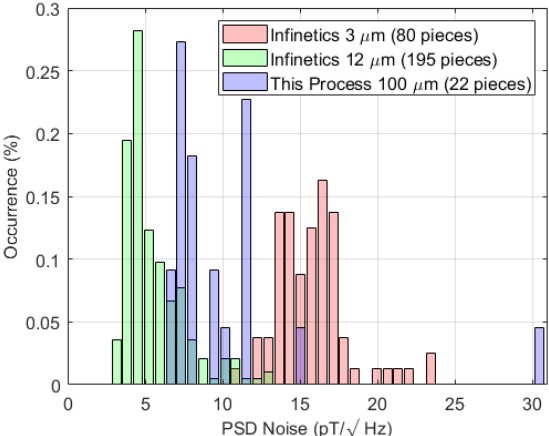

**Figure 19: Histogram of ring-core noise in pT Hz$^{-1/2}$ at 1 Hz. The current process yields 6–11 pT Hz$^{-1/2}$ at 1 Hz noise performance compared to ~4 for the Infinetics 12 μm ring-cores and ~16 for the 3 μm Infinetics ring-cores.**

The number of ring-cores produced by the current process was lower (22 pieces) than for the Infinetics 3 μm (80 pieces) or 12 μm

(195 pieces) processes available for comparison. The current process primarily yielded ring-cores with a noise of 6–11 pT Hz$^{-1/2}$ at 1 Hz. The most recent batch of ten ring-cores had a median magnetic noise of 7.5 pT Hz$^{-1/2}$ at 1 Hz with a standard deviation of 2.5 pT Hz$^{-1/2}$. A subset of six ring-cores underwent a nominal spaceflight survival qualification testing which consisted of exposure to vacuum, vibration testing, and eight thermal cycles from -60 °C to +75 °C. The performance of the six ring-cores was tested before and after this testing and was not negatively impacted.

**7 Summary and Conclusions**

Despite their ubiquitous use, the parameters that control the intrinsic magnetic noise of the ferromagnetic element remain poorly understood and no published process can reproduce the performance of the widely used low-noise Infinetics S1000 ring-core. We described a manufacturing approach that can reliably produce a S1000 compatible fluxgate ring-core with a noise of 6–11 pT Hz$^{-1/2}$ at 1 Hz. The ring-core comprised a custom 6-81 permalloy, cold-rolled to 100 μm, coated with an insulator, spirally wound onto

a circular bobbin, and heat treated in a reducing atmosphere. The manufacturing process described here is unlikely to be either a local or global optimum in the ring-core parameter space. However, it does reproduce performance comparable to the legacy S1000 ring-core that underpins many modern instruments. As a result, our team has established a process that can provide an ongoing supply of scientifically useful low-noise ring-cores which will ensure an ongoing supply of high-quality science-grade fluxgate magnetometer instruments.

**8 Future Work**

Ongoing research into optimising this process and new low-noise materials and sensors is being conducted with the goal of ~1 pT Hz$^{-1/2}$ at 1 Hz noise levels. Research is in progress to explore parameters such as: alternative ring-core geometries (including racetracks and parallel rods), the constituents and makeup of the permalloy, the use of geometries such as wire rather than strips, the permalloy thickness, the necessity of work-hardening the ferromagnetic material before heat treating, the impact of 100%

versus 5% hydrogen in the heat treatment atmosphere, the rate of heating to the dwell temperature, the temperature and duration of the dwell, the rate of cooling from dwell to the disordering temperature, and the rate of cooling from the disordering temperature



to room temperature. The high-noise outliers reported here suggest that there are some as-yet poorly controlled manufacturing variables with a significant negative impact on the ring-core noise. An obvious potential way to improve the repeatability of the manufacturing process would be an improved way to affix the permalloy strip onto the bobbin. Experiments to evaluate other ring-core construction techniques such as solid permalloy washers rather than spiral wound tape are in progress.

## 5   Code and Data Availability

Data and source code used in the creation of this paper can be accessed by contacting the authors.

## Author Contributions

D. M. Miles wrote the manuscript with contributions from all authors. The ring-cores presented here were developed at the University of Alberta under a contract led by I. R. Mann from the Canadian Space Agency. D. M. Miles developed and built the experimental apparatus to characterise the new ring-cores. B. B. Narod primarily completed the literature review into the ring-core physics and developed the heat treatment presented here. J. R. Bennest provided guidance on the ring-core manufacturing process. M. Ciurzynski and D. Barona designed, constructed and operated the heat-treatment system. A. Kale primarily characterised ring-cores. M. R. Lessard completed preliminary tests of the ring-core manufacturing process. D. K. Milling primarily provided guidance on testing and characterising ring-core performance. J. Larson built, assembled, and tested infrastructure at the University of Iowa used to characterise and document ring-cores for this manuscript.

## Competing Interests

B. B. Narod operated Narod Geophysics Ltd., which manufactured fluxgate magnetometers until the company ceased production operation in 2008. J. R. Bennest operated Bennest Enterprises Ltd., which manufactured a variety of custom scientific instruments and equipment including fluxgate magnetometers until the company ceased operations in 2018.

## Acknowledgements

Work on the project was supported by the Canadian Space Agency under contract 9F063-140909/006/MTB_PT6_Ring-cores. D. M. Miles was subsequently supported by faculty start-up funding from the University of Iowa. I. R. Mann is supported by a Discovery Grant from Canadian NSERC. The authors wish to thank Richard Dvorsky, Michael Webb, Christian Hansen, and Spencer Kuhl for developing, manufacturing, and documenting ring-cores shown in this manuscript.

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
