# Peer review of "Low-Noise Permalloy Ring-Cores for Fluxgate Magnetometers"

_Geoscientific Instrumentation, Methods and Data Systems, 2019_

## Referee Comment (RC1) · Anonymous Referee #1 · 18 Jun 2019

The authors present the production of 6-81 permalloy based low-noise ring-cores essential for high-quality, science-grade fluxgate instrumentation. This is very important work for the community since especially the North American market dramatically lacks manufactures for this type of magnetic sensing element due to the fact that Infinetics Inc. stopped the production in 1996. The paper is very well written, clearly understandable and an appropriate number of citations is included.

There are two general remarks:

1) The authors miss to mention the ring-core development activities in Europe. This fact gets especially obvious with the citation of Carr et al., 2005 as a potential use of Infinetics ring-cores. This is not correct. The outboard sensor was produced by Ultra Electronics (Kellock et al., 1996 and Carr et al., 2007) in the UK and the inboard sensor

was developed by the Technical University in Braunschweig, Germany (Fornacon et al. 1999 and Auster et al. 2008). In addition, there has also been a very relevant ring-core production at the DTU in Kopenhagen (e.g. Nielsen et al., 1999).

Auster, H.U., Glassmeier, K.H., Magnes, W., Aydogar, O., Baumjohann, W., Constantinescu, D., Fischer, D., Fornacon, K.H., Georgescu, E., Harvey, P., Hillenmaier, O., Kroth, R., Ludlam, M., Narita, Y., Nakamura, R., Okrafka, K., Plaschke, F., Richter, I., Schwarzl, H., Stoll, B., Valavanoglou, A., Wiedemann, M., 2008. The THEMIS Fluxgate Magnetometer. Space Sci Rev 141, 235–264. https://doi.org/10.1007/s11214-008-9365-9

Carr et al., A Magnetometer For The Solar Orbiter Mission, The Second Solar Orbiter Workshop, 16-20 October 2006, Athens, Greece, ESA Publ. Div. (2007) https://www.researchgate.net/publication/41625175_A_Magnetometer_For_The_Solar_Orbiter_Mission

Fornacon, K.-H., Auster, H.U., Georgescu, E., Baumjohann, W., Glassmeier, K.-H., Haerendel, G., Rustenbach, J., Dunlop, M., 1999. The magnetic field experiment onboard Equator-S and its scientific possibilities. Annales Geophysicae 17, 1521–1527. https://doi.org/10.1007/s00585-999-1521-3

Nielsen, O.V., Brauer, P., Primdahl, F., Risbo, T., Jørgensen, J.L., Boe, C., Deyerler, M., Bauereisen, S., 1997. A high-precision triaxial fluxgate sensor for space applications: layout and choice of materials. Sensors and Actuators A: Physical 59, 168–176. https://doi.org/10.1016/S0924-4247(97)80169-0

2) The authors primarily focus on the noise level of the ring-cores at 1 Hz and forget about the offset stability with ring-core temperature. This is a very important parameter for a "science grade" fluxgate instrumentation. This shortcoming should be discussed in Section 6 (Performance of the Ring-Cores) and/or Section 8 (Future Work).

---

## Referee Comment (RC2) · Anonymous Referee #2 · 4 Jul 2019

Review on "Low-Noise Permalloy Ring-Cores for Fluxgate Magnetometers" by David M. Miles et al.

**General Comments**

The manuscript is devoted to reproduction and detailed documenting of the manufacturing process of ring-core fluxgate sensors based on Ni81.3-Mo6-Fe soft magnetic alloy. This low-noise material was successfully used as a sensor magnetic core in many science-grade fluxgate magnetometers with excellent offset stability and noise level. The renewed techniques and developed equipment is a good base for further experiments with soft magnetic materials for achieving the goal of creating extremely low-noise fluxgate sensors.
The paper is well prepared and easy to understand.
The authors tried to reproduce the sensor developed by Gordon et al. (1968) and produced by Infinetics Inc. as a S1000 ring-cores till 1996. However, Gordon et al. (1968) paper inspired other research groups (for instance, in Germany and former USSR) to experiment with similar magnetic alloys and they reported even better noise level of fluxgate sensors with smaller cores.
Yu.V. Afanassiev, one of the leaders of these researches in former USSR, collected many useful information on fluxgate magnetometers in his book "Ferrozondovye pribory (Fluxgate instruments)" issued in Russian in 1986 and in English translation as a part of the book "Fluxgate Magnetometers for Space Research" by G. Musmann and Yu. Afanassiev in 2010. In the next section, along with other comments, I will try to give some citations from the latter book, which are highly relevant to the paper content.

**Specific Comments**

**p. 2, l. 13-14,** *"A preferred ferromagnetic material used in fluxgate sensors is 6-81 permalloy containing 6% molybdenum, 81.3% nickel, and the remainder iron."*

Perhaps, the soft magnetic alloy with the similar composition (Ni81Mo6Fe) is produced by Aperam company:
https://www.aperam.com/specialty-alloys-offer/product-categories
https://www.aperam.com/sites/default/files/documents/2018-01/Strip_SUPERMIMPHY_LLS.pdf

If this alloy is commercially available, what are the main reasons to prepare custom-made samples?

In the paper Afanas'ev et al. (1977) noises of the five alloys (79%-83% of Nickel) with low saturation magnetostriction coefficients and different heat treatment modes were analyzed in the rod-cores (130x2.8x0.1 mm) fluxgates. Three of them 83NiV (Ni82.5-

84.2V3.8-4.2Fe), 81NiMo/special (Ni80.5-81.7Mo4.7-5.2Ti2.5-3.2Fe) and 82Ni6Mo (Ni81.0-81.8Mo5.8-6.2Fe) had similar noise levels.

Other examples of the 81-6 Mo permalloy use in fluxgate sensors are given in:

1) Auster et al. (2008):

"The ring-cores used for Themis have been developed by Karl Heinz Fornacon in Germany for more than 20 years (Müller et al. 1998). The main design goals have always been low noise and offset stability over a wide temperature range and period of time. Material selection and preparation as well as a proper thermal treatment are the key steps to achieve the performance parameters required for the Themis mission. The applied soft-magnetic material, a 13Fe-81Ni-6Mo alloy, is rolled to a foil of 20 μm thickness. Ribbons with a width of 2 mm are cut and 7 turns of it are wound on a bobbin made from Inconel.
...
The selection of the ring-cores relies on an extended test procedure. After winding the excitation coil directly onto the ring core bobbins the noise of each ring core is measured before and after a specific aging process which consists of ultra sonic treatment, vibration, and temperature cycling. The sensor noise at 1 Hz of a ring core with a diameter of 13 mm is typically less than 5 pT/√Hz as shown in Fig. 2."

2) Musmann (2010), p. 140-142:

"...This core is made by winding a thin Permalloy 82NiMo tape of 0.02 mm gauge and 1.5 mm width, polished, degreased and coated with a heat resistant insulation over a metallic bobbin and annealing them together. … The six turns are wound into the groove with constant tension where the end is being fixed to the preceding turn one by spot welding."

The core diameter was 13.2 mm. The sensor droved at the excitation frequency f=12.5 kHz had noise level < 5 pT Hz$^{-1/2}$ at 1 Hz with amplitude of the excitation field 2-2.5 kA m$^{-1}$ and < 3 pT Hz$^{-1/2}$ with 3-3.3 kA m$^{-1}$. These average results were obtained by testing over 100 specimens.

**p. 9, l. 7-9,**

*"The assembled fluxgate ring-cores were heat treated to produce high-permeability, low-coercivity, and repeatable re-magnetization properties in the ferromagnetic material that the authors hypothesize produced a relatively stress-free crystalline structure and therefore low magnetic noise. This approach was guided by the theory of the origin of fluxgate magnetic noise developed in Narod (2014). The heat treatment was intended to develop the largest possible grains in the given thickness of the permalloy foil without developing undesirable fabric where the easy axes directions are misaligned with respect to the desired magnetizing direction (e.g., Major and Martin, 1970; Odani, 1964)."*

Gorobei and Gorobei (1981) experimentally showed that a low-temperature (800 °C) annealing of Nickel-Molybdenum alloys yields lower noise level than a high-temperature (1000, 1150 °C) annealing recommended in Gordon et al. (1968).

Müller et al. (1998) had confirmed this result for alloys Ni81Mo6Fe and Ni80Mo6Fe and showed that the minimal noise level achieved for fine-grained (grain < 13 μm) samples.

**p. 6, l. 10-19,**

*"The bobbins were manufactured from Inconel x750 that was selected as being non-magnetic and providing high rigidity even at the elevated temperatures of the heat treatment required to optimize the magnetic properties of the ring-core. However, Inconel x750 was a imperfect match to the permalloy sense element in terms of linear thermal expansion (12.6 ppm °C$^{-1}$ for Inconel x750, and estimated to be about 11.6 ppm °C$^{-1}$*

*for 6-81 permalloy).* *Properties for the Inconel x750 were taken from the Special Metals Group of Companies data sheet, Unified Numbering System for Metals and Alloys, reference UNS N07750. The immediate impact of the thermal mismatch was differential expansion during the heat treatment leading to a loose fit of the permalloy strip on the bobbin in the final ring-core assembly. The differential expansion may also have enhanced the magnetic noise by introducing mechanical stress during the heat treatment and if the final ring-core assembly was operated over a wide temperature range. This effect has not yet been investigated in detail. For future designs, alternative bobbin materials that are a closer thermal match are being explored."*

One of the possible solutions of this problem (matching linear thermal expansion coefficients of a bobbin and a ferromagnetic core) were proposed earlier in Musmann (2010), p. 107, last paragraph:

"… for example the alloy NiMo+X…(non-magnetic alloy with a high specific resistance $\rho=1.5 – 1.5 \ \Omega mm^2 m^{-1}$, a high melting temperature (1350 °C) and linear expansion coefficient close to that of permalloy ($\alpha=12 \cdot 10^{-6} \ K^{-1}$))"

Probably the same alloy was mentioned in Afanassiev et al. (1980), where authors proposed to use for core bobbins (instead of Inconel x750) the high resistance Nickel-Molybdenum alloy ($Mo_{22-24}Cr_{2.5-2.9}Al_{1.8-2.2}Ni_{bal}$, HM23ХЮ in Russian notation) due to its 2-3 times lower magnetic susceptibility and better matching the linear temperature coefficient with that of permalloy.

**p. 6, l. 21,**
*"A custom 4 kg ingot of 6-81 permalloy was created using a vacuum arc furnace to create a 50–50 alloy of molybdenum and nickel and then melting in the remaining constituents in a conventional furnace."*

Gordon et al. (1968) used "induction-melting electrolytic iron and nickel in a hydrogen atmosphere, adding molybdenum, and pouring in helium."
What was the reason of using slightly different approach for ingot preparation?

**p. 8, l. 1-6,**
*"The Permalloy strips were coated with magnesium oxide (an electrical insulator) to prevent the formation of spot welds between layers when the strip was attached by electrical discharge welding, and to prevent the tightly-wound layers from fusing with each other during heat treatments. This insulator was created using an established process (Bill Billingsley Sr, personal communication) from milk of magnesia, MgOH diluted with water to reduce its viscosity and form a consistent thin layer. The strip was dipped, hung up, and allowed to air dry (Figure 8a). During the heat treatment the milk of magnesia residue (Figure 8b) formed a thin but robust layer of magnesium oxide MgO that electrically isolates each layer and minimises eddy currents."*

The other possible insulation coatings were mentioned in Musmann (2010), p. 107, third paragraph from the bottom:

"Before winding, the tape must be degreased and covered by a heat-resistant insulator. These operations are often performed during the winding process. For insulation, electrolysis is applied and the tape is coated with different suspensions.
During electrolysis, an insulating substance being in a suspension state is applied to the tape surface electrostatically. Use is made of different suspensions: silicon dioxide in acetone, magnesium oxide in carbon tetrachloride, ammonium oxide in methyl alcohol and others. The tape in the electrolysis is transported at 1 to 3 m/min. The covering thickness is regulated by changing the speed, the suspension concentration, and the applied voltage."

**p. 8, l. 13-20,**
*"The end of the permalloy strip was spot welded to the bottom of the channel cut into the outer circumference of the bobbin. The strip was cut to length such that the start and end of the strip were aligned. ... The spiral winding was terminated by scraping away a small amount of the oxide layer (Figure 9b) and spot-welding the end of the strip down to the layer immediately underneath (Figure 9c)"*

The method of fixing a soft magnetic strip to a bobbin is given in Musmann (2010), p. 155, Section "Ring Core Design and Manufacturing":
"...After etching of the cutting zone the tape is wound (normally 6-12 windings) and fixed on a bobbin. One end of the tape may be fixed on the bobbin using point electrowelding. Then the tape is wound under stress and the end of outer winding is electrowelded to the previous one"

Was the permalloy strip wound with constant tension?

**Technical corrections**

**p. 16, l. 5-11**

What were a sample frequency of the analyzed time series and the lowest frequency of the power spectral density estimations?

**p. 17, Fig. 19.** Histograms for materials "Infinetics 3 μm..." and "This Process 100 μm…" are poorly distinguishable in a grayscale image.

**References**

Yu. V. Afanas'ev, V. N. Gorobei, K. D. Mart'yanova, B. Ya. Pluchek, V. V. Sosnin, and T. I. Shcherbakova, "Low-noise iron-nickel alloys for cores of ferromagnetic probes and magnetic modulators," *Measurement Techniques*, vol. 20, no. 10, pp. 1491–1494, Oct. 1977.

H. U. Auster *et al.*, "The THEMIS Fluxgate Magnetometer," *Space Science Reviews*, vol. 141, no. 1–4, pp. 235–264, Dec. 2008.

"Fluxgate magnetometers for Space Research", ed. G. Musmann, 2010, 280 p.

V. N. Gorobei and N. N. Gorobei, "Thermal-activation approach to the study of noise in cores of ferromagnetic probes," *Measurement Techniques*, vol. 24, no. 12, pp. 1085–1088, Dec. 1981.

M. Müller, T. Lederer, K. H. Fornacon, and R. Schäfer, "Grain structure, coercivity and high-frequency noise in soft magnetic Fe81Ni6Mo alloys," Journal of Magnetism and Magnetic Materials, vol. 177, pp. 231–232, 1998.

USSR Copyright (Inventors) Certificate 771580, Yu.V. Afanassiev, V.N. Gorobei, V.P. Porfirov, B.Ya. Pluchek, T.I. Scherbakova, V.Ya. Agornik, 1980, [in Russian], http://patents.su/3-771580-ustrojjstvo-dlya-izmereniya-parametrov-magnitnogo-polya.html

---

## Author Comment (AC1) · 6 Aug 2019

We thank the referee for the constructive comments which we have incorporated into the manuscript. Referee #1 raised several questions and issues which we address below; the referee's comments are in plain text our responses in *italics* and any content added to or changed in the manuscript are in *"quoted italics*

The authors present the production of 6-81 permalloy based low-noise ring-cores essential for high-quality, science-grade fluxgate instrumentation. This is very important work for the community since especially the North American market dramatically lacks manufactures for this type of magnetic sensing element due to the fact that Infinetics Inc. stopped the production in 1996. The paper is very well written, clearly understandable and an appropriate number of citations is included.

There are two general remarks:

1) The authors miss to mention the ring-core development activities in Europe. This fact gets especially obvious with the citation of Carr et al., 2005 as a potential use of Infinetics ring-cores. This is not correct. The outboard sensor was produced by Ultra Electronics (Kellock et al., 1996 and Carr et al., 2007) in the UK and the inboard sensor was developed by the Technical University in Braunschweig, Germany (Fornacon et al. 1999 and Auster et al. 2008). In addition, there has also been a very relevant ring-core production at the DTU in Kopenhagen (e.g. Nielsen et al., 1999).

*We agree with the reviewer that the outboard sensor on Double Star was manufactured by Ultra Electronics. However, there is significant, albeit circumstantial, evidence that it was manufactured from Infinetics S1000 ring-cores. Around 1993 there was a management buy-out at Ultra Electronics that appears to have included Domain Magnetics/Dowty Aerospace who are known to have held significant stock of Infinetics ring-cores. The Double Star sensor has comparable geometry to the Cassiope/e-POP sensor suggesting that its ring-cores are at least geometrically similar to the Infinetics S1000 and have similar noise performance of <7 pT/√Hz at 1 Hz (Carr, 2006). Finally, Ultra Electronics has stated the ring-cores were manufactured in the USA (Carr, personal communication, 2013).*

*Text added: "In some cases, the providence of the ring-cores is complicated and difficult to know for certain. For example, the outboard sensor on Double Star was manufactured by Ultra Electronics (Carr et al., 2005), the inboard sensor having been developed at the Technical University in Braunschweig, Germany (Auster et al., 2008; Fornacon et al., 1999). However, there is significant, albeit circumstantial, evidence that the Ultra Electronics sensor was manufactured from Infinetics S1000 ring-cores. Around 1993 there was a management buy-out at Ultra Electronics that appears to have included Domain Magnetics/Dowty Aerospace who are known to have held significant stock of Infinetics ring-cores. The Double Star sensor has comparable geometry to the Cassiope/e-POP (Wallis et al., 2015) sensor suggesting that its ring-cores are at least geometrically similar to the Infinetics S1000. Ultra Electronics sensors also have similar noise performance of  <7 pT/√Hz at 1 Hz (Carr et al., 2007; Kellock et al., 1996) . Finally, Ultra Electronics has stated the ring-cores were manufactured in the USA (Carr, 2013 personal communication)."*

*The pointer to the ring-core production at DTU is well made and was raised by both reviewers. We have added a description of these activities, and other European work in the new Section 7 Discussion.*

*Text added: - See new Section 7*

Auster, H.U., Glassmeier, K.H., Magnes, W., Aydogar, O., Baumjohann, W., Constantinescu, D., Fischer, D., Fornacon, K.H., Georgescu, E., Harvey, P., Hillenmaier, O., Kroth, R., Ludlam, M., Narita, Y., Nakamura, R., Okrafka, K., Plaschke, F., Richter, I., Schwarzl, H., Stoll, B., Valavanoglou, A., Wiedemann, M., 2008. The THEMIS Fluxgate Magnetometer. Space Sci Rev 141, 235–264. https://doi.org/10.1007/s11214- 008-9365-9

Carr et al., A Magnetometer For The Solar Orbiter Mission, The Second Solar Orbiter Workshop, 16-20 October 2006, Athens, Greece, ESA Publ. Div. (2007) https://www.researchgate.net/publication/41625175_A_Magnetometer_For_The_Solar_Orbiter_Mission

Fornacon, K.-H., Auster, H.U., Georgescu, E., Baumjohann, W., Glassmeier, K.-H., Haerendel, G., Rustenbach, J., Dunlop, M., 1999. The magnetic field experiment onboard Equator-S and its scientific possibilities. Annales Geophysicae 17, 1521–1527. https://doi.org/10.1007/s00585-999-1521-3

Nielsen, O.V., Brauer, P., Primdahl, F., Risbo, T., Jørgensen, J.L., Boe, C., Deyerler, M., Bauereisen, S., 1997. A high-precision triaxial fluxgate sensor for space applications: layout and choice of materials. Sensors and Actuators A: Physical 59, 168–176. https://doi.org/10.1016/S0924-4247(97)80169-0

2) The authors primarily focus on the noise level of the ring-cores at 1 Hz and forget about the offset stability with ring-core temperature. This is a very important parameter for a "science grade" fluxgate instrumentation. This shortcoming should be discussed in Section 6 (Performance of the Ring-Cores) and/or Section 8 (Future Work).

*We agree – investigations of offset stability would be particularly valuable given the limited existing literature on the topic.*

*Text added: "The presented work has focused on manufacturing ring-cores that meet the commonly accepted noise level of < 10 pT / √Hz at 1 Hz. Future work will need to include investigation of other performance metrics including changes in the offset stability, gain, and noise of the ring-core with temperature."*

---

## Author Comment (AC2) · 6 Aug 2019

We thank the referee for the constructive comments which we have incorporated into the manuscript. Referee #2 raised several questions and issues which we address below; the referee's comments are in plain text our responses in *italics* and any content added to or changed in the manuscript are in *"quoted italics*

General Comments

The manuscript is devoted to reproduction and detailed documenting of the manufacturing process of ring-core fluxgate sensors based on Ni81.3-Mo6-Fe soft magnetic alloy. This low-noise material was successfully used as a sensor magnetic core in many science-grade fluxgate magnetometers with excellent offset stability and noise level. The renewed techniques and developed equipment is a good base for further experiments with soft magnetic materials for achieving the goal of creating extremely low-noise fluxgate sensors.

The paper is well prepared and easy to understand.

The authors tried to reproduce the sensor developed by Gordon et al. (1968) and produced by Infinetics Inc. as a S1000 ring-cores till 1996. However, Gordon et al. (1968) paper inspired other research groups (for instance, in Germany and former USSR) to experiment with similar magnetic alloys and they reported even better noise level of fluxgate sensors with smaller cores.

Yu.V. Afanassiev, one of the leaders of these researches in former USSR, collected many useful information on fluxgate magnetometers in his book "Ferrozondovye pribory (Fluxgate instruments)" issued in Russian in 1986 and in English translation as a part of the book "Fluxgate Magnetometers for Space Research" by G. Musmann and Yu. Afanassiev in 2010. In the next section, along with other comments, I will try to give some citations from the latter book, which are highly relevant to the paper content.

*Thank-you for these references – we have incorporated them into the manuscript as described below. We have also introduced a new discussion section at the end of the manuscript which provides a broader international context of fluxgate ring-core development.*

*Text added:*

*"7 Discussion*

*Our work towards developing a well-defined and reliable method for manufacturing low-noise fluxgate sensors for space-physics and geophysics uses has generally followed on the efforts undertaken by Naval Ordinance Laboratory in the 1960's (Gordon et al., 1968), the commonalities being the use of 6-81 Mo permalloy as the principal material, and its processing by cold deformation and specific heat treatment for the development of large grain structure. However, several other methodologies making use of different materials have also been used for the making of low-noise fluxgate sensors, with significant success.*

*Beginning in 1984 with the experimental works of Shirae (1984) and Narod et al (1985) amorphous, high-cobalt alloys and processes were developed to create fluxgate sensors for both ground and space-based uses. By ways of example, In the 1990's Otto Neilson and others at Danish Technical University (Nielsen et al., 1995) used an amorphous alloy to create the CSC ring-core fluxgate magnetometer aboard the Orsted satellite. Also beginning in the 1990's Luis Benyoseph of the National Observatory of Brazil has undertaken a long term experimental effort to improve amorphous alloys for fluxgate sensors (Benyosef, 1996; Benyosef et al., 1995, 2008). More recently Lajos Varga at the Hungarian Academy of Sciences developed amorphous materials for magnetometry (L Varga, personal communication, 2013).*

*Rapidly quenched amorphous alloys are limited by both their minimum and maximum thicknesses, as compared with crystalline permalloys. Also, the rapid cooling creates an anisotropy in the cooling direction across the thickness. Both properties may limit their possible performance in magnetometers. Their low Curie temperatures, 200C or lower, may impact their thermal stability (Acuña, 2002).*

*In the 1990's investigators in Braunschweig, Berlin and Dresden developed a low-noise permalloy, again having 6.0% molybdenum and about 81% nickel (Müller et al., 1998). The processing of this material was significantly different from our present material. From conversations with K.H. Fornacon (K.H. Fornacon, personal communication, 2017) we understand that their heat treatments were performed in a furnace that did not permit rapid insertion of specimens into the hot zone and that specimens were required to warm up slowly, as the furnace was heated. Such a heating curve would eliminate all possibility of grain growth by primary recrystallization occurring, leaving all grain growth to occur via secondary recrystallization (Pfeifer and Radeloff, 1980). It has been believed for a long time that secondary recrystallization is undesirable for the development of magnetic properties (Odani, 1964). Their overall result was that their best material for fluxgate sensors had the smallest grain size, about 10 microns.*

*Significant work with various permalloys has been published in the Russian language literature (e.g., Afanas'ev, 1986; Afanas'ev et al., 1977; Afanassiev et al., 1980; Musmann, 2010). However, limited details of the heat treatments or grain sizes from the work completed around the 1970s are available in the English language literature.*

*If one takes as common knowledge that low coercivity, high initial permeability are desirable traits for sensor materials, then there should be merit is seeking materials of very small grain. Herzer (1990, 1992) found that such desirable properties occur at both ends of the grain size spectrum, that is for grain sizes much smaller than 1 micron, or larger than 20 microns. The small grain end of the spectrum gets into the so-called nanocrystalline materials, while the large grain end is the parameter space we have chosen to examine.*

*The process described herein has been developed so that eventually all the essential steps (melting, rolling, cutting, assembling, heat-treating, calibrating, and testing) should be achievable in-house and in small quantities. This capability is intended to enable future studies to probe the many different design choices that may affect fluxgate performance (metallurgical composition, alloying process, homogenisation, reduction method, work-hardening, machining and heat affect, geometry, heat-treatment profile, etc). The authors would like to encourage other researchers to publish as many of these details as is practical to facilitate meaningful comparison of results between different cores manufactured by different groups using different processes to eventually expose all the critical parameters that enable low-noise, highly-stable materials for fluxgate sensors."*

Specific Comments

p. 2, l. 13-14, "A preferred ferromagnetic material used in fluxgate sensors is 6-81 permalloy containing 6% molybdenum, 81.3% nickel, and the remainder iron."

Perhaps, the soft magnetic alloy with the similar composition (Ni81Mo6Fe) is produced by Aperam company:

https://www.aperam.com/specialty-alloys-offer/product-categories

https://www.aperam.com/sites/default/files/documents/2018-01/Strip_SUPERMIMPHY_LLS.pdf

If this alloy is commercially available, what are the main reasons to prepare custom-made samples?

*Thank-you for the reference which we have included in the manuscript. Our general experience with commercial alloy providers is that the minimum purchase requirements (often ~100 kg) make them expensive for small quantity development work – especially if one intends to explore different alloy regimes. Our impression is that the long-duration heat treatments typically required for 6% Molybdenum alloys limit their commercial usage. We also had access to the Canadian government CANMET lab, now in Hamilton, which offered flexible melting options at reasonable price.*

*It was also CANMET's staff's recommendation to mix the Ni and Mo together before going for the full 4 kg melt and we followed their recommendations. We now understand that it is possible to dissolve Molybdenum into liquid Nickel at ~1450 C providing a simple method for melting small quantities of 6-81 alloy.*

*Custom-making our own alloy also allows us to control grain development. Our goal is to affordably manufacture small ingots that we can then heat treat further, and at length, to control both grain development and homogeneity.*

*We have updated the relevant section of the manuscript to capture these details.*

*Text now reads:*

*"A custom 4 kg ingot of 6-81 permalloy was created at the Canadian government CANMET lab. Following the suggestion of CANMET staff, a vacuum arc furnace was used to create a 50–50 alloy of molybdenum and nickel and then melting in the remaining constituents in a conventional furnace. Subsequent work has shown that it is possible to dissolve molybdenum into nickel at ~1450 °C providing a simple method to manufacture small quantities of 6-81 permalloy in a laboratory setting. Some 6-81 alloys are now available from commercial sources (e.g. SUPERMIMPHY LLS manufactured by Aperame). However, minimum purchase requirements (often ~100 kg) can make commercial procurement expensive for small quantity development. Custom manufacturing of the 6-81 alloy also ensures that its heat treatment history, which controls both grain development and homogeneity, are well documented."*

In the paper Afanas'ev et al. (1977) noises of the five alloys (79%-83% of Nickel) with low saturation magnetostriction coefficients and different heat treatment modes were analyzed in the rod-cores

(130x2.8x0.1 mm) fluxgates. Three of them 83NiV (Ni82.5-84.2V3.8-4.2Fe), 81NiMo/special (Ni80.5-81.7Mo4.7-5.2Ti2.5-3.2Fe) and 82Ni6Mo (Ni81.0-81.8Mo5.8-6.2Fe) had similar noise levels.

Other examples of the 81-6 Mo permalloy use in fluxgate sensors are given in:

1) Auster et al. (2008):

"The ring-cores used for Themis have been developed by Karl Heinz Fornacon in Germany for more than 20 years (Müller et al. 1998). The main design goals have always been low noise and offset stability over a wide temperature range and period of time. Material selection and preparation as well as a proper thermal treatment are the key steps to achieve the performance parameters required for the Themis mission. The applied soft-magnetic material, a 13Fe-81Ni-6Mo alloy, is rolled to a foil of 20 µm thickness. Ribbons with a width of 2 mm are cut and 7 turns of it are wound on a bobbin made from Inconel.

...

The selection of the ring-cores relies on an extended test procedure. After winding the excitation coil directly onto the ring core bobbins the noise of each ring core is measured before and after a specific aging process which consists of ultra sonic treatment, vibration, and temperature cycling. The sensor noise at 1 Hz of a ring core with a diameter of 13 mm is typically less than 5 pT/√Hz as shown in Fig. 2."

2) Musmann (2010), p. 140-142:

"...This core is made by winding a thin Permalloy 82NiMo tape of 0.02 mm gauge and 1.5 mm width, polished, degreased and coated with a heat resistant insulation over a metallic bobbin and annealing them together. ... The six turns are wound into the groove with constant tension where the end is being fixed to the preceding turn one by spot welding."

The core diameter was 13.2 mm. The sensor droved at the excitation frequency f=12.5 kHz had noise level < 5 pT Hz-1/2 at 1 Hz with amplitude of the excitation field 2-2.5 kA m-1 and < 3 pT Hz-1/2 with 3-3.3 kA m-1. These average results were obtained by testing over 100 specimens.

*Thank-you for the references – we have included them into the manuscript.*

*Text added: "Two other groups are known to have constructed fluxgate sensors from 6-81 permalloy. The Themis (Auster et al., 2008) ring-cores result from years of research in Germany (Müller et al., 1998) and achieved noise better than 5 pT / √Hz using a 20 µm foil. Musmann (2010) describe similar noise performance and also used a 20 µm foil."*

p. 9, l. 7-9,

"The assembled fluxgate ring-cores were heat treated to produce high-permeability, low-coercivity, and repeatable re-magnetization properties in the ferromagnetic material that the authors hypothesize produced a relatively stress-free crystalline structure and therefore low magnetic noise. This approach was guided by the theory of the origin of fluxgate magnetic noise developed in Narod (2014). The heat treatment was intended to develop the largest possible grains in the given thickness of the permalloy

foil without developing undesirable fabric where the easy axes directions are misaligned with respect to the desired magnetizing direction (e.g., Major and Martin, 1970; Odani, 1964)."

Gorobei and Gorobei (1981) experimentally showed that a low-temperature (800 °C) annealing of Nickel-Molybdenum alloys yields lower noise level than a high- temperature (1000, 1150 °C) annealing recommended in Gordon et al. (1968).

Müller et al.  (1998) had confirmed this result for alloys Ni81Mo6Fe and Ni80Mo6Fe and showed that the minimal noise level achieved for fine-grained (grain < 13 μm) samples.

*Fair point – clearly there is more to be understood here before we can say exactly what material process causes what noise results. We have acknowledge these results for fine-grained materials in the manuscript.*

*Text added: "In contrast, Gorobei and Gorobei (1981) showed experimentally that a low-temperature (800 °C) annealing of Nickel-Molybdenum alloys yields lower noise level than a high- temperature (1000, 1150 °C) annealing recommended in Gordon et al. (1968). Further study is needed to understand what material process causes what noise result. The discrepancy between Couderchon et al. (1989) and Müller et al. (1998) is striking. An 800 °C treatment that is purely secondary recrystallization may well be optimal for smaller grain materials, but this would need to be investigated."*

p. 6, l. 10-19,

"The bobbins were manufactured from Inconel x750 that was  selected as  being  non-magnetic and providing high rigidity even at the elevated temperatures of the heat treatment required to optimize the  magnetic properties of the ring-core. However, Inconel x750 was a  imperfect match to the permalloy sense element in  terms of linear thermal expansion (12.6 ppm °C- 1 for Inconel x750, and estimated to be about 11.6 ppm °C-1 for 6-81 permalloy). Properties for the Inconel x750 were taken from the Special Metals Group of Companies data sheet, Unified Numbering System for Metals and Alloys, reference UNS N07750. The immediate impact of the thermal mismatch was differential expansion during the heat treatment leading to a loose fit of the permalloy strip on the bobbin in the final ring-core assembly. The differential expansion may also  have  enhanced the magnetic noise by introducing mechanical stress during the heat treatment and if the final ring- core assembly was operated over a wide temperature range. This effect has not yet been investigated in detail. For future designs, alternative bobbin materials that are a closer thermal match are being explored. "

One of the possible solutions of this problem (matching linear thermal expansion coefficients of a bobbin and a ferromagnetic core) were proposed earlier in Musmann (2010), p. 107, last paragraph:

"…  for example the alloy NiMo+X…(non-magnetic alloy with a  high specific resistance ρ=1.5 –  1.5 Ωmm2m-1, a high melting temperature (1350 °C) and linear expansion coefficient close to that of permalloy (α=12·10-6 K- 1))"

Probably the same alloy was mentioned in Afanassiev et al.  (1980),  where  authors proposed to use for core bobbins (instead of Inconel x750) the high resistance Nickel-Molybdenum alloy (Mo22-24Cr2.5-

2.9Al1.8-2.2Nibal, HM23ХЮ in Russian notation) due to its 2-3 times lower magnetic susceptibility and better matching the linear temperature coefficient with that of permalloy.

*Thank you for the suggestion and the references. We have included them into the manuscript.*

*Test now reads: "For future designs, alternative bobbin materials that are a closer thermal match are being explored. Potential candidates include Haynes alloy B3 and the non-magnetic NiMo+X alloy class suggested by Musmann (2010) and potentially referenced by Afanassiev et al., (1980). Both have low chromium content, which is desirable as the chromium becomes volatile when heat-treated at higher temperatures."*

p. 6, l. 21,

"A custom 4 kg ingot of 6-81 permalloy was created using a vacuum arc furnace to create a 50–50 alloy of molybdenum and nickel and then melting in the remaining constituents in a conventional furnace."

Gordon et al. (1968) used "induction-melting electrolytic iron and nickel in a hydrogen atmosphere, adding molybdenum, and pouring in helium."

What was the reason of using slightly different approach for ingot preparation?

*This approach was suggested by staff at the Canadian government CANMET lab where the melt was conducted – the text has been revised to make this clear.*

*Text now reads: "A custom 4 kg ingot of 6-81 permalloy was created at the Canadian government CANMET lab. Following the suggestion of CANMET staff a vacuum arc furnace was used to create a 50–50 alloy of molybdenum and nickel and then melting in the remaining constituents in a conventional furnace"*

p. 8, l. 1-6,

"The Permalloy strips were coated with magnesium oxide (an electrical insulator) to prevent the formation of spot welds between layers when the strip was attached by electrical discharge welding, and to prevent the tightly-wound layers from fusing with each other during heat treatments. This insulator was created using an established process (Bill Billingsley Sr, personal communication) from milk of magnesia, MgOH diluted with water to reduce its viscosity and form a consistent thin layer. The strip was dipped, hung up, and allowed to air dry (Figure 8a). During the heat treatment the milk of magnesia residue (Figure 8b) formed a thin but robust layer of magnesium oxide MgO that electrically isolates each layer and minimises eddy currents. "

The other possible insulation coatings were mentioned in Musmann (2010),

p. 107, third paragraph from the bottom:

"Before winding, the tape must be degreased and covered by a heat-resistant insulator. These operations are often performed during the winding process. For insulation, electrolysis is applied and the tape is coated with different suspensions.

During electrolysis, an insulating substance being in a suspension state is applied to the tape surface electrostatically. Use is made of different suspensions: silicon dioxide in acetone, magnesium oxide in carbon tetrachloride, ammonium oxide in methyl alcohol and others. The tape in the electrolysis is transported at 1 to 3 m/min. The covering thickness is regulated by changing the speed, the suspension concentration, and the applied voltage."

*Thank-you for the reference – we have incorporated it into the manuscript.*

*Text added: "Other insulating coatings capable and application methods are possible, as described by Musmann (2010), the main requirement being that they can survive the heat treatment in a Hydrogen atmosphere."*

 p. 8, l. 13-20,

"The end of the permalloy strip was spot welded to the bottom of the channel cut into the outer circumference of the bobbin. The strip was cut to length such that the start and end of the strip were aligned The spiral winding was terminated by scraping away a small amount of the oxide layer (Figure 9b) and spot-welding the end of the strip down to the layer immediately underneath (Figure 9c)"

The method of fixing a soft magnetic strip to a bobbin is given in Musmann (2010), p. 155, Section "Ring Core Design and Manufacturing"

"After etching of the cutting zone the tape is wound (normally 6-12 windings) and fixed on a bobbin. One end of the tape may be fixed on the bobbin using point electrowelding. Then the tape is wound under stress and the end of outer winding is electrowelded to the previous one"

*Noted – we have acknowledged this in the manuscript.*

*Text now reads: "A similar method of assembling rings is described in Musmann (2010)."*

Was the permalloy strip wound with constant tension?

*Not specifically - the foil was wound by hand using a simple jig so the tension was not finely controlled. We have acknowledged this in the manuscript.*

*Text now reads: "In this case, the foil was wound by hand using a simple jig and the tension was not finely controlled"*

Technical corrections p. 16, l. 5-11

What were a sample frequency of the analyzed time series and the lowest frequency of the power spectral density estimations?

*The timeseries was digitized at 100 sps and the described Fourier transform estimated the power spectral density down to ~0.1 Hz.*

*Text now reads: "The solenoid was then switched off and 20 minutes of 100 sps magnetically quiet data were recorded and processed to produce power spectral density (PSD) using a unit-correct quantitative implementation of Welch's method (Heinzel et al., 2002; Welch, 1967) with a 2048 point Fast Fourier Transform (FFT), a 1536 point overlap, and a Hann window function giving a power spectral density estimation down to ~0.1 Hz."*

p. 17, Fig. 19. Histograms for materials "Infinetics 3 μm…" and "This Process 100 μm…" are poorly distinguishable in a grayscale image.

*Fair point – we have updated the figure to include hash patterns that should show even when rendered in greyscale.*

[Figure]

References

Yu. V. Afanas'ev, V. N. Gorobei, K. D. Mart'yanova, B. Ya. Pluchek, V. V. Sosnin, and T. I. Shcherbakova, "Low-noise iron-nickel alloys for cores of ferromagnetic probes and magnetic modulators," Measurement Techniques, vol. 20, no. 10, pp. 1491–1494, Oct. 1977.

H. U. Auster et al., "The THEMIS Fluxgate Magnetometer," Space Science Reviews, vol. 141, no. 1–4, pp. 235–264, Dec. 2008.

"Fluxgate magnetometers for Space Research", ed. G. Musmann, 2010, 280 p.

V. N. Gorobei and N. N. Gorobei, "Thermal-activation approach to the study of noise in cores of ferromagnetic probes," Measurement Techniques, vol. 24, no. 12, pp. 1085–1088, Dec. 1981.

M. Müller, T. Lederer, K. H. Fornacon, and R. Schäfer, "Grain structure, coercivity and high-frequency noise in soft magnetic Fe81Ni6Mo alloys," Journal of Magnetism and Magnetic Materials, vol. 177, pp. 231–232, 1998.

USSR Copyright (Inventors) Certificate 771580, Yu.V. Afanassiev, V.N. Gorobei, V.P. Porfirov, B.Ya. Pluchek, T.I. Scherbakova, V.Ya. Agornik, 1980, [in Russian], http://patents.su/3-771580-ustrojjstvo-dlya-izmereniya- parametrov-magnitnogo-polya.html